**COMMENT**

# Thinking clearly about misinformation

Li Qian Tay [1,2✉], Stephan Lewandowsky [1,3,4], Mark J. Hurlstone[5], Tim Kurz [1] & Ullrich K. H. Ecker [1,6]

There is concern that many ills in Western societies are caused by misinformation. Some researchers argue that misinformation is merely a symptom, not a cause. This is a false dichotomy, and research should differentiate between dimensions of misinformation in these evaluations.

In Western societies, misinformation concern is at an all-time high. Recently, however, debate has ensued regarding the level of concern that is warranted. Some researchers note the potential for misinformation to incur significant costs on individuals and societies, and call for interventions to reduce misinformation susceptibility and impacts[1,2]. Others warn against alarmist narratives, arguing that misinformation exerts only limited influence over beliefs and behaviours. This view proposes that problematic behaviours, such as vaccine hesitancy, are caused by systemic socio-economic and psycho-social issues, and thus calls for interventions to target those societal issues rather than misinformation creation and consumption, which represent only symptoms of these deeper issues[3]. Similarly, assuming low prevalence of misinformation, researchers have argued that interventions should focus on increasing trust in factual information[4].

A principled way to resolve these contradicting analyses is needed, to better inform policies and minimize the risk of enshrining a problematic status quo or investing resources to address a perhaps negligible problem. We argue that misinformation has had clear impacts; that depending on individual and contextual factors, it can be both a symptom and a cause; and that its multidimensionality (e.g. topic, type, and depth of dissemination) ought to be more fully considered when making such evaluations.

## A call for causal clarity

Societal issues can shape individuals' beliefs and produce problematic behaviours. Behaviours such as vaccine hesitancy and climate-change denial have been facilitated by factors such as populism, inequality, disenfranchisement, political polarization, and the concentration of media ownership[5]. These factors are amplified by low institutional trust, which is a wicked problem because even if many institutions are generally trustworthy, some politicians, scientists, media outlets, and corporations have engaged in unethical behaviours that do warrant scepticism.

Yet, even if the misinformation problem is symptomatic of such deeper issues, this does not negate the fact that symptoms can cause outcomes of their own. To illustrate: A factor such as inequality might increase the symptom of misinformation susceptibility, while misinformation itself might cause belief changes or behaviours (unrelated to inequality) in a causal chain; alternatively, a factor such as polarization or institutional distrust might causally affect misinformation susceptibility, which in turn might further entrench polarization or distrust in a vicious cycle.

[1] School of Psychological Science, University of Western Australia, Perth, WA, Australia. [2] School of Medicine and Psychology, Australian National University, Canberra, ACT, Australia. [3] School of Psychological Science, University of Bristol, Bristol, UK. [4] Department of Psychology, University of Potsdam, Potsdam, Germany. [5] Department of Psychology, Lancaster University, Lancaster, UK. [6] Public Policy Institute, University of Western Australia, Perth, WA, Australia. ✉email: li.tay@anu.edu.au

A counterfactual perspective can provide further clarification: causation is essentially the difference between a world in which a putative cause is present and a counterfactual world in which all is equal except for the absence of the cause. Thus, if misinformation were merely a symptom, then nothing in the world would change if all misinformation were to disappear. This is clearly implausible. Observational and experimental studies have demonstrated that misinformation can causally alter beliefs and behaviours[1,6], even though measurement of misinformation impacts is often impeded by ethical considerations (e.g. exposing individuals to potentially harmful misinformation) or lack of access to relevant data (e.g. historical or transnational data; data from social-media platforms or closed channels such as offline communications and encrypted chat applications). Indeed, in a counterfactual world without any misinformation, false beliefs could only emerge via spontaneous generation. Such spontaneous generation is not uncommon (e.g. stereotypes and superstitions can result from social processes or illusory correlations). However, it would be inadequate as an all-encompassing explanation for the spread of false beliefs that go beyond individuals' immediate experiences or observations. For example, the widespread false belief that the mumps-measles-rubella (MMR) vaccine causes autism would be unlikely to gain traction had fraudulent MMR-vaccine research not received high-profile media coverage.

Critically, the counterfactual perspective can account for multi-causality. Consider a situation in which an individual is influenced by a claim that a vaccine is harmful. Both the misinformation and the existing susceptibility of the individual (e.g. low trust in science) are causal factors, if, without either, the individual would not have been misinformed to the same extent (e.g. formed a weaker misconception). Whether the misinformation or the existing susceptibility is a better explanation then depends on their relative prevalence and the probability of sufficiency. For example, in case of a fire breaking out after an individual lights a match, match-lighting may be a better explanation for the fire than the presence of oxygen, because oxygen is more prevalent than match-lighting and the individual lighting the match should have anticipated the presence of oxygen (such analyses are used in legal reasoning to determine damages)[7]. Thus, even if institutional mistrust can partially explain some individuals' tendency to be affected by vaccine misinformation (alongside other individual-specific factors such as perceived plausibility, worldview congruence, utility for behaviour justification, etc.), it does not absolve the causal responsibility of misinformants, nor negate the potential effects of vaccine misinformation on public health.

One way to capture the complexity of such causal networks is through directed acyclic graphs, as shown in Fig. 1. This approach can also illustrate how existing research has focused on specific direct effects within limited timeframes, often neglecting more indirect causal factors and potentially important context variables. For example, the existing misinformation literature is biased towards a liberal-democratic, Western framework and has largely overlooked the potential influence of environmental context factors such as state capacity and the presence of ethnic conflicts or historical grievances, which may co-determine misinformation impacts.

In sum, it is important to avoid a false dichotomy. The key question is not whether misinformation is better framed as a symptom or a cause of social issues, but rather under what conditions one framing is more appropriate than the other. In doing so, there is a selection of misinformation dimensions that should be considered to appropriately recognize misinformation heterogeneity.

## Recognizing heterogeneity

Objectively and easily identifiable misinformation, typically referred to as fake news, represents only a small portion of the average person's information diet in Western societies[8]. However, in our view, (1) the misinformation problem should not be considered negligible because a subset of obvious misinformation has low prevalence, and (2) it is unreasonable to expect all types of misinformation to always have strong effects on all outcomes. Some studies will find misinformation has minimal effects, others may suggest the opposite[9]—as a generalization, both characterizations will be inaccurate unless qualified with explicit recognition of heterogeneity.

To this end, we direct attention to three key dimensions of misinformation—topic, type, and depth—that will influence its real-world reach and impact. The first dimension, topic, refers

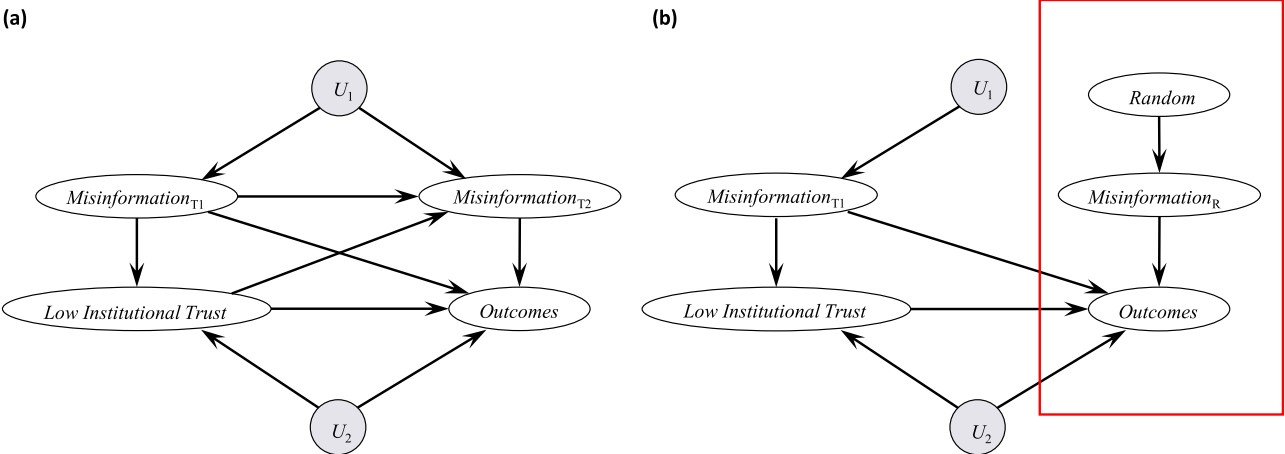

**(a)**

**(b)**

**Fig. 1 Directed acyclic graphs illustrating causal networks of misinformation effects. a** Directed acyclic graphs are graphical causal models characterized by nodes representing variables and edges representing direct causal effects. In the example, both low institutional trust and misinformation can cause outcomes such as vaccine hesitancy. Additionally, low trust and misinformation can have cross-lagged effects (e.g. low trust at Time 1 causes more misinformation at Time 2), and there are likely other relevant factors ($U_1$ and $U_2$; e.g. technological and economic conditions, state capacity, or specific events); **b** Research leveraging randomization, on average, controls for spurious factors and allows causal identification for a subset of misinformation (*Misinformation*$_R$). However, many studies tend to focus on a limited timescale, estimating only specific direct effects and not total effects (e.g. nodes and arrows within the red box, where the effects of prior misinformation and other context factors ($U_2$) might not be captured).

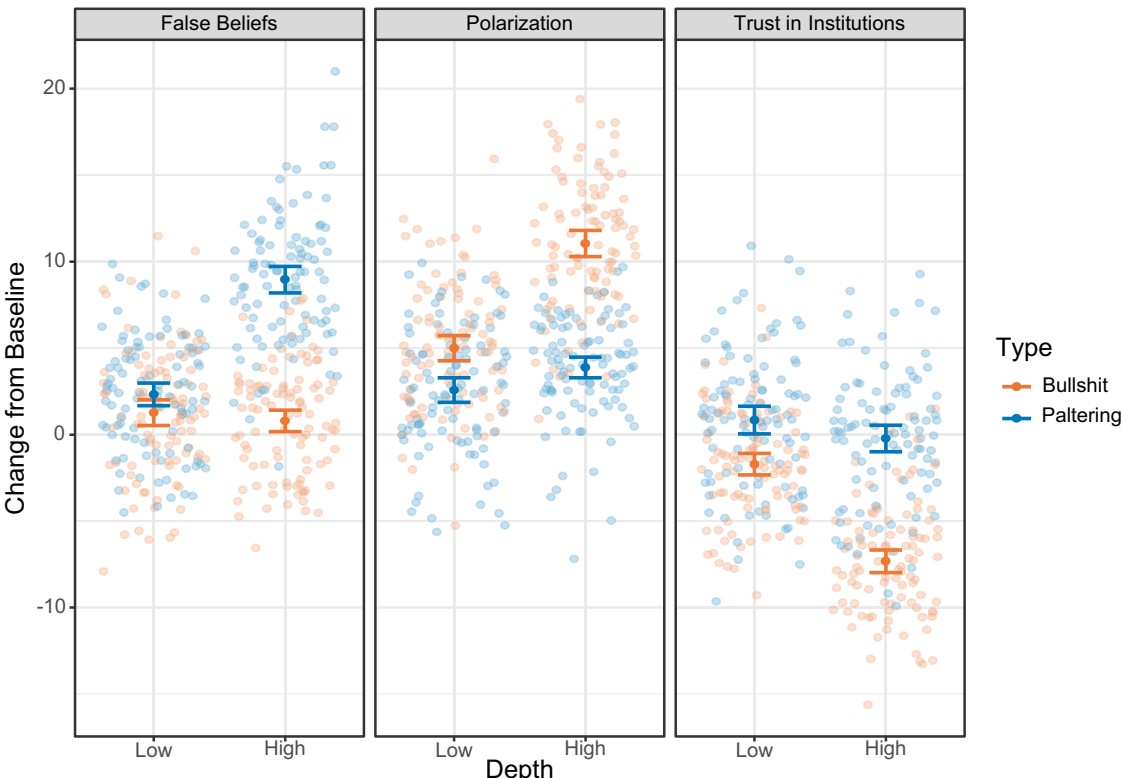

**Fig. 2 Potential misinformation effects across types, depths, and outcomes.** Graphical illustration of some potential misinformation effects. Plotted data and error bars are hypothetical and on an arbitrary scale, to illustrate that different misinformation types at various depths can have different impacts across outcomes. For example, bullshit, even at high depth, may have minimal effects on beliefs, but may drive polarization and mistrust (even if identified as misleading). By contrast, paltering may affect beliefs without affecting trust (because fewer individuals will identify the misinformation).

simply to the subject matter of the information. For instance, individuals in Country A will be impacted more by misinformation about a specific situation (e.g. an election) in Country A than similar misinformation regarding Country B.

Second, with regards to type, we follow McCright and Dunlap[10] in distinguishing between truthiness (misleading information that simply feels true), systemic lies (carefully crafted misinformation advancing ideological interests), bullshit (persuasive misinformation used opportunistically with total disregard for evidence), and shock-and-chaos (large volumes of content that aim to confuse or fatigue). Note that not all information captured in this framework will need to be literally false; for example, some information that is "truthy" or part of a shock-and-chaos approach might not be objectively false or even falsifiable (e.g. in a conflict situation, the narrative that the adversary is scared). Similarly, the selective, slanted, or miscontextualized presentation of true information can be used to mislead, an approach sometimes referred to as paltering. Supplementary Table 1 applies this categorization to a selection of real-world misinformation. Considering this diversity, it becomes clear that much misinformation is advanced—intentionally or unintentionally—by sources that would typically not be categorized as dubious by researchers estimating misinformation prevalence. For example, Grinberg and colleagues[8] focussed exclusively on websites known to publish fabricated stories. This leaves subtler types of misinformation outside of researchers' tallies; if these neglected types are considered, misinformation will be found to occupy a greater portion of the information landscape.

The third key dimension, depth, relates to both distribution and repetition. The distributional aspect refers to whether the misinformation is dispersed haphazardly (e.g. individual social-media posts or headlines) or if content is systematically bundled

and/or targeted (e.g. an organized disinformation campaign; a revisionist history curriculum). The repetitional aspect relates to the well-known finding that repeated and thus familiar information is more likely judged to be true regardless of veracity[1]. Misinformation depth is important to consider because pieces of misinformation can have compound impacts[11]. Much like a river can be fed from multiple tributaries, multiple information sources can contribute to the same false narrative. This narrative gist can then be shared by downstream distributaries, which can include individuals never exposed to any initial misinformation, or news organizations that would never refer to the original low-quality sources. In this manner, misleading narratives can infiltrate mainstream news coverage and influence public discourse (e.g. conspiratorial claims influencing public debate during "Pizzagate"). Thus, assessing prevalence without accounting for narrative gist will systematically underestimate the scale of the misinformation problem.

Critically, potential outcomes can differ across misinformation types and depths, and can be undesirable even if the misinformation is identified. For example, a Republican correctly identifying bullshit from a Democrat might have lowered opinions of Democrats (or vice versa), which can fuel polarization even without any direct impact on beliefs. Even the discourse surrounding misinformation itself can have negative effects (e.g. erode satisfaction with electoral democracy[12]). Figure 2 presents an idealized illustration of some potential misinformation impacts across types, depths, and outcomes.

A final point is that active forces can drive misinformation consumption. For instance, a vaccine-hesitant individual seeking vaccine information will encounter more vaccine misinformation than someone who is incidentally exposed. Moreover, vulnerable individuals may be targeted with misinformation tailored to their

---

**Box 1 | Recommendations for future research**

First, a focus shift in misinformation-intervention evaluation is recommended. To illustrate: One of the most popular paradigms presents participants with large sets of true and false claims, with the difference in truth or belief ratings between the two taken as a measure of discernment. This paradigm limits studies to short-format misinformation (e.g. headlines, tweets), as tasking participants to engage with lengthier misinformation (e.g. articles, videos) in large sets can be impractical. This favours light-touch interventions that might not address persuasive misinformation at higher depth, even though such misinformation could be more impactful.

Second, future research should make more use of observational causal-inference strategies. Regardless of how realistic or incentivized laboratory-based measures can be, it remains true that many factors are not manipulable due to ethical or feasibility considerations. For example, researchers have used the positioning of cable-TV channels (which varies randomly across localities in the US) in instrumental-variable analyses showing that exposure to unreliable news sources reduced social-distancing behaviours during the early stages of the COVID-19 pandemic[13]. Both observational and laboratory work should move towards a more global, comparative perspective, given that existing studies have mostly focussed on Western societies, and it remains unclear whether results generalize to other contexts.

Finally, as an integrative account of false beliefs is lacking, another promising direction is to borrow from the broader cognitive-science literature. For instance, cognitive research has shown that individuals preferentially rely on gist representations of quintessential meanings[11]. Future research attempting to delineate the evolution of narrative gist at a societal level might therefore benefit from first examining gist processing at the individual level. Cognitive models of decision making could also be used to explore misinformation impact beyond observable outcomes. For example, evidence-accumulation models could be used to decompose choice and response-time data into cognitively interpretable parameters (e.g. response boundaries represent the varying levels of evidence individuals require to make decisions and could be interpreted as caution).

---

psychological vulnerabilities. If this has the potential to cause harm (to the individual or the public good), then it should be of concern, even if overall consumption is low or if such misinformation only strengthens pre-existing attitudes. Caution is therefore needed when making general claims of prevalence and (lack of) impacts based on limited data.

## Conclusions and recommendations

Taken together, a clear implication of our discussion is that the standard paradigms as well as the limited (typically Western) contexts used for evaluating the impacts of misinformation and misinformation interventions are likely insufficient. Some recommendations for changes to current research practice in the field are provided in Box 1. We have argued that the evaluation of misinformation impacts is an important, but complex, research question, particularly in the current era of rising geopolitical tensions and rapid technological change. We hope that the current Comment will contribute to increasingly nuanced debates about the impact of misinformation and potential interventions.

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

## Acknowledgements

L.Q.T. received funding from a University Postgraduate Award and a UWA International Fees Scholarship. U.K.H.E. was supported by an Australian Research Council grant (FT190100708). S.L. acknowledges financial support from the European Research Council (ERC Advanced Grant 101020961 PRODEMINFO), the Humboldt Foundation through a research award, the Volkswagen Foundation (grant "Reclaiming individual autonomy and democratic discourse online: How to rebalance human and algorithmic decision making"), and the European Commission (Horizon 2020 grant 101094752 SoMe4Dem). The funders had no role in the preparation of the manuscript or decision to publish.

## Author contributions

L.Q.T. drafted the original manuscript. S.L., M.J.H., T.K., and U.K.H.E. provided critical revisions.

## Competing interests

The authors declare no competing interests.
