## [Peer Review File · Communications Psychology]

4th Dec 23

Dear Mr Tay,

Your Comment titled "Thinking Clearly About Misinformation" has now been seen by 1 Referee, who also commented on earlier versions. Their comments appear below. In light of their advice, I am delighted to say that we are happy, in principle, to publish it in Communications Psychology under a Creative Commons 'CC BY' open access license.

We will not send your revised paper for further review if, in the editors' judgment, all comments on the present version have been addressed. If the revised paper is in Communications Psychology format, in an accessible style, and of appropriate length, we shall accept it for publication immediately. I have attached an edited version of your manuscript, and ask you to attend to each comment in detail.

We therefore invite you to revise your paper one last time to address a list of editorial requests. At the same time we ask that you edit your manuscript to comply with our format requirements and to maximise the accessibility and therefore the impact of your work.

EDITORIAL REQUESTS:

I have attached a marked-up version of your manuscript to guide these final revisions. Please note that the format dictates a limit of 10 references. We are strictly limited in the number of references by the nature of the piece. We can as an exception grant an extra 2-3 references, but this requirement is strict. In the marked-up version of the manuscript, I commented on the appropriate use of references.

Moreover, in the original decision letter you received, we specified based on the comments made by Reviewer #3 that: "[...] most research articles in this domain arise from a few western / Global North / anglophone countries. This limitation of the primary research, which naturally affects its presentation in the present piece, needs to be acknowledged from the start and feed into the presentation of the solutions or actionable points that arise from the research". This has not yet been sufficiently incorporated. Acknowledgment of the limitation of the data must be integrated into the first paragraphs and revisited in the outlook sections (and the box with recommendations for further research) as a precondition of acceptance.

Please review our specific editorial comments and requests regarding your manuscript in the attached "Editorial Requests Table". Please outline your response to each request in the right-hand column. Please upload the completed table with your manuscript files as a Related Manuscript file.

* Please review the changes in the attached copy of your manuscript, which has been edited for style, and address the comments and queries I have added. If using Word, please use the 'track changes' feature to make the process of accepting your manuscript more efficient.

* Please check whether your manuscript contains third-party images, such as figures from the literature, stock photos, clip art or commercial satellite and map data. If any of the display items in

your manuscript (figures, tables, boxes or movies) include images that are the same as, or are adaptations of, previously published images, please fill in the [Third Party Rights Table](http://www.nature.com/licenceforms/snl/thirdpartyrights-table.doc), and return to us when you submit your revised manuscript. This information will enable us to obtain the necessary rights to re-use such material. If we are unable to obtain the necessary rights to use or adapt any of the material that you wish to use, we will contact you to discuss alternative options.

* Communications Psychology uses a transparent peer review system. On author request, confidential information and data can be removed from the published reviewer reports and rebuttal letters prior to publication. If you are concerned about the release of confidential data, please let us know specifically what information you would like to have removed. Please note that we cannot incorporate redactions for any other reasons.

*If you have not done so already, please alert me to any related manuscripts from your group that are under consideration or in press at other journals, or are being written up for submission to other journals (see www.nature.com/authors/editorial_policies/duplicate.html for details).

FORMATTING GUIDELINES:

You will find a complete list of formatting requirements following this link:

<https://www.nature.com/documents/commsj-style-formatting-checklist-comment.pdf>

Please use the checklist to prepare your manuscript for final submission. In the following, I also highlight some issues of particular importance.

** Length

The ideal length for Comment article in Communications Psychology is 1,500 words. We have some flexibility, however, but please ensure that your text does not exceed 1,800 words.

** Figures

Please remove all figures from the main text and upload them individually, one figure per file. To ensure the swift processing of your paper please provide the highest quality, vector format, versions of your images (.ai, .eps, .psd) where available. Text and labelling should be in a separate layer to enable editing during the production process. If vector files are not available then please supply the figures in whichever format they were compiled in and not saved as flat .jpeg or .TIFF files. If your artwork contains any photographic images, please ensure these are at least 300 dpi.

* Figures should be simple and informative — multi-part figures are best avoided.

* References

References appear as superscript Arabic numerals, in order of mention. The reference list mentions references in the numerical order in which they are mentioned in the main text. If a reference is cited more than once, the same number is used throughout the text and the reference receives a single entry in the reference list.

Only papers that have been published or accepted by a named publication should be in the reference list (preprints and citations of datasets are also permitted). Unpublished/Submitted research should not be included in the reference list; it should only be mentioned briefly and

parenthetically in the main text. Note that no major arguments should rely on unpublished research.

Published conference abstracts and URLs for websites should be cited parenthetically in the text, not in the reference list.

Footnotes are not used.

* Competing interests

Please include a "Competing interests" statement after the References. Note that we ask authors to declare both financial and non-financial competing interests. For more details, see

<https://www.nature.com/authors/policies/competing.html>. If you have no financial or non-financial competing interests, please state so: "The authors declare no competing interests."

SUBMISSION INFORMATION:

* If you wish, you may also submit a visually arresting image, together with a concise legend, for consideration as a 'Hero Image' on our homepage. The file should be 1400x400 pixels and should be uploaded as 'Related Manuscript File'. In addition to our home page, we may also use this image (with credit) in other journal-specific promotional material.

In order to accept your paper, we require the following:

* A cover letter describing your response to our editorial requests.

* The final version of your text as a Word or TeX/LaTeX file, with any tables prepared using the Table menu in Word or the table environment in TeX/LaTeX and using the 'track changes' feature in Word.

* Production-quality versions of all figures, supplied as separate files. Photographic images should be 300 dpi in RGB format (.jpg, TIFF or native Photoshop format) and any labels/scale bars included in a separate layer from the image. Line art, graphs and schemes should be vector format (.ai, .eps, .pdf); Adobe Illustrator files are preferred and will minimize production time. Any chemical structures or schemes contained within figures should additionally be supplied as separate Chemdraw (.cdx) files.

At acceptance, the corresponding author will be required to complete an Open Access Licence to Publish on behalf of all authors, declare that all required third-party permissions have been obtained.

Please note that your paper cannot be sent for typesetting to our production team until we have received this information; **therefore, please ensure that you have this ready when submitting the final version of your manuscript.**

ORCID

Communications Psychology is committed to improving transparency in authorship. As part of our efforts in this direction, we are now requesting that all authors identified as 'corresponding author'

create and link their Open Researcher and Contributor Identifier (ORCID) with their account on the Manuscript Tracking System (MTS) prior to acceptance. ORCID helps the scientific community achieve unambiguous attribution of all scholarly contributions. For more information please visit <http://www.springernature.com/orcid>

For all corresponding authors listed on the manuscript, please follow the instructions in the link below to link your ORCID to your account on our MTS before submitting the final version of the manuscript. If you do not yet have an ORCID you will be able to create one in minutes.

IMPORTANT: All authors identified as 'corresponding author' on the manuscript must follow these instructions. Non-corresponding authors do not have to link their ORCIDs but are encouraged to do so. Please note that it will not be possible to add/modify ORCIDs at proof. Thus, if they wish to have their ORCID added to the paper they must also follow the above procedure prior to acceptance.

To support ORCID's aims, we only allow a single ORCID identifier to be attached to one account. If you have any issues attaching an ORCID identifier to your MTS account, please contact the [Platform Support Helpdesk](http://platformsupport.nature.com/).

[link redacted]

We hope to hear from you within two weeks; please let us know if the process may take longer.

Best wishes
Marike

Marike Schiffer, PhD
Chief Editor
Communications Psychology

REVIEWERS' COMMENTS:

Reviewer #1 (Remarks to the Author):

We believe the manuscript is much improved as a result of these revisions. There is however still some room for improvement. Below, we raise minor points, and suggest recent articles that the authors could discuss to strengthen some arguments. If word limit is a problem, we encourage the Editor to be lenient, as the discussion of these recent articles could benefit the manuscript.

Minor points:

Discussing “Allen, J., Watts, D. J., & Rand, D. (2023). Quantifying the Impact of Misinformation and Vaccine-Skeptical Content on Facebook” in the current article could strengthen the argument that biased or clumsy reporting by reliable news source probably matters more than false news posted by unreliable websites.

We also believe that discussing this article <https://arxiv.org/abs/2308.06459> would strengthen the “misinformation narrative” part of the article (that is still relatively weak in our opinion given that it is mostly supported by verbal arguments instead of actual data).

“A final point is that active forces can drive misinformation consumption. For instance, a vaccine-hesitant individual seeking vaccine information will encounter more vaccine misinformation than someone who is incidentally exposed. Moreover, vulnerable individuals may be targeted with misinformation tailored to their psychological vulnerabilities. If this has the potential to cause harm (to the individual or the public good), then it should be of concern, even if overall consumption is low. Caution is therefore needed when making general claims of prevalence and (lack of) impacts based on limited data.”

The self-selection process described here (well documented in the literature) also implies that misinformation is unlikely to create, *de novo*, new problematic behaviors or attitudes. The authors could address this, we believe, by extending “even if overall consumption is low” to “even if overall consumption is low and even if such misinformation only strengthen pre-existing attitudes.” Because even if misinformation does not convince pro-vax people to become anti-vax, it can still be harmful if it strengthens pre-existing anti-vax attitudes (or if it weakens pro-vax attitudes).

Related to the previous comments, this recent paper on the effect of repetition is a great example of how misinformation can have detrimental effects even if it doesn’t influence the whole population but only subgroups: Pillai, R. M., Kim, E., & Fazio, L. K. (2023). All the President’s lies: Repeated false claims and public opinion. *Public opinion quarterly*, 87(3), 764-802.” It is also an important reminder that harmful misinformation often comes from the top rather than from low quality websites.

We still believe that domain-level estimation of misinformation prevalence (such as studies relying on NewsGuard ratings) mostly looks at the prevalence of low-quality information (i.e., produced with low journalistic standards), that this category is much broader than misinformation or fake news (given that only about half of news shared by these domains is actually false), and thus that these studies likely overestimate the prevalence of misinformation. But it is an open debate, and we can agree to disagree! (The Authors are probably right though about the importance of subtler forms of misinformation in mainstream media, but there is little empirical evidence to support this to date, especially outside of the arguably unusual case of the US).